# lhCLIP reveals the in vivo RNA–RNA interactions recognized by hnRNPK

**Yuanlang Hu[1,2,3], Tao Hao[1,4¤], Hanwen Yu[5], Wenbin Miao[2], Yi Zheng[2], Weihua Tao[1,4], Jingshen Zhuang[1], Jichang Wang[5]\*, Yujuan Fan[2]\*, Shiqi Jia[1,4,6]\***

**1** Department of General Surgery, The First Affiliated Hospital, Jinan University, Guangzhou, People's Republic of China, **2** Ministry of Science and Education, University of Chinese Academy of Sciences-Shenzhen Hospital, Shenzhen, People's Republic of China, **3** College of basic medical sciences, Three Gorges University, Yichang, People's Republic of China, **4** The Guangdong-Hong Kong-Macao Joint University Laboratory of Metabolic and Molecular Medicine, Jinan University, Guangzhou, People's Republic of China, **5** Key Laboratory for Stem Cells and Tissue Engineering (Sun Yat-sen University), Ministry of Education, Guangzhou, People's Republic of China, **6** Key Lab of Guangzhou Basic and Translational Research of Pan-vascular Diseases, Guangzhou, People's Republic of China

¤ Current address: Department of Colorectal Hernia Surgery, Binzhou Medical University Hospital, Binzhou, People's Republic of China.

\* wangjch53@mail.sysu.edu.cn (JW); yjfan530@163.com (YF); shiqijia@jnu.edu.cn (SJ)

**Data Availability Statement:** The raw data of lhCLIP have been deposited in NCBI's Sequence Read Archive (SRA) with the accession number PRJNA838401.

## Abstract

RNA-RNA interactions play a crucial role in regulating gene expression and various biological processes, but identifying these interactions on a transcriptomic scale remains a challenge. To address this, we have developed a new biochemical technique called pCp-biotin labelled RNA hybrid and ultraviolet crosslinking and immunoprecipitation (lhCLIP) that enables the transcriptome-wide identification of intra- and intermolecular RNA-RNA interactions mediated by a specific RNA-binding protein (RBP). Using lhCLIP, we have uncovered a diverse landscape of intermolecular RNA interactions recognized by hnRNPK in human cells, involving all major classes of noncoding RNAs (ncRNAs) and mRNA. Notably, hnRNPK selectively binds with snRNA U4, U11, and U12, and shapes the secondary structure of these snRNAs, which may impact RNA splicing. Our study demonstrates the potential of lhCLIP as a user-friendly and widely applicable method for discovering RNA-RNA interactions mediated by a particular protein of interest and provides a valuable tool for further investigating the role of RBPs in gene expression and biological processes.

## Author summary

RBPs play a crucial role in post-transcriptional regulation of RNAs by modulating RNA-RNA interactions, encompassing processes like pre-mRNA splicing and mRNA degradation. Existing techniques for exploring RNA interactions associated with specific RBPs require some troublesome experimental procedures, including the use of radioisotopes to visualize RBP-RNA complexes and the introduction of exogenous expression of the target protein. In this context, we introduce lhCLIP, a novel method designed for capturing RNA-RNA interactions mediated by a specific protein. Diverging from other approaches reliant on radioisotopes, lhCLIP eliminates the need for their use. Moreover, its

**Funding:** This work was supported by the China Postdoctoral Science Foundation (2022M711337 to YH), National Natural Science Foundation of China (31970856 to SJ), Clinical Frontier Technology Program of the First Affiliated Hospital of Jinan University, China (JNU1AF-CFTP-2022-a01236 to SJ), Key Research Projects of the University of Chinese Academy of Sciences-Shenzhen Hospital (HRF-2021004 to YH), National Natural Science Foundation of China (31871444 to JW), National Natural Science Foundation of China (81900702 to WT), National Key Research and Development Program of China (2018YFA0107003 to JW), Guangdong Provincial Natural Science Foundation (2021A1515010537 to JW), Science and Technology Program of Guangzhou: Key Lab of Guangzhou Basic and Translational Research of Pan-vascular Diseases (202201020042). The funders had no role in study design, data collection and analysis, decision to publish, or preparation of the manuscript.

**Competing interests:** The authors have declared that no competing interests exist.

heightened sensitivity enables the capture of RNA-RNA interactions mediated by endogenously expressed RBPs, making it an intuitive and competitive option for researchers. We utilized lhCLIP to investigate RNA hybrids that are directly bound by hnRNPK, unveiling a complex array of intra- and intermolecular RNA interactions with the potential to exert influence over RNA splicing.

## Introduction

Genomic studies have revealed that over 85% of the human genome is transcribed into RNA [1,2]. However, in mammalian cells, only 2–7% of the total transcripts by mass are messenger RNA (mRNA), while the majority are noncoding RNAs (ncRNAs) [3]. Despite their prevalence, the functions of most ncRNAs remain unknown. Important functions of several ncRNAs were discovered by studying RNA–RNA interactions [4]. These interactions have been found to play crucial roles in various cellular processes, including transcription, pre-mRNA splicing, protein translation, and mRNA degradation [4].

However, identifying RNA interactions on a transcriptomic scale remains a significant challenge. Recently, several techniques have been developed to map the RNA interactome, including MARIO (Mapping RNA interactome in vivo) [5], LIGR-seq (LIGation of interacting RNA followed by high-throughput sequencing) [6] and RIC-seq (RNA in situ conformation sequencing) [7]. These methods profile the topology of all RNA-binding proteins (RBPs)-mediated RNA-RNA interactions. However, there is still a need for strategies to identify RNA interactions associated with specific RBPs. CLASH (crosslinking, ligation, and sequencing of hybrids) [8,9] and hiCLIP (RNA hybrid and individual-nucleotide resolution ultraviolet cross-linking and immunoprecipitation) [10] have been developed to detect RNA duplexes bound by specific RBPs. However, both techniques require introduction of exogenous expression of the target protein and can only provide chimeric read coverage of approximately 2%. Recently, the hiCLIP atlas of duplexes bound by Staufen1(STAU1) has been extended by approximately10-fold through the development of bespoke computational methods which were apply to existing data [10]. Additionally, a novel method called captured RICseq (CRIC-seq) has been developed to profile the RNA interactome arranged by a single RBP [11]. However, the use of radioisotopes to visualize RBP-RNA complexes in hiCLIP has proven to be a challenge for inexperienced experimenters. Additionally, CRIC-seq lacks experimental procedures for visualizing RBP-RNA complexes.

Here, we present lhCLIP (pCp-biotin labelled RNA hybrid and ultraviolet crosslinking and immunoprecipitation), a novel method for capturing RNA-RNA interactions mediated by a specific protein. lhCLIP incorporates pCp-biotin (biotinylated cytidine (bis) phosphate) to label the RBP-RNA complexes, enabling easy detection of the biotin moiety using streptavidin conjugated with horseradish peroxidase (HRP) in a chemiluminescent assay. Comparing to other methods, lhCLIP does not require the radioisotopes, furthermore, its heightened sensitivity enables the capture of RNA-RNA interactions mediated by endogenously expressed RBPs, making it a user-friendly and competitive option for researchers. We utilized lhCLIP to investigate RNA hybrids that are directly bound by hnRNPK, a DNA/RNA-binding protein [12]. hnRNPK contains 3 K-homologous (KH) domains that recognize nucleic acids and facilitate the protein's binding to RNA and DNA [13]. hnRNPK function as an enhancer or repressor of gene transcription and a docking platform for the signal transduction proteins. It interacts with splicing factors and regulates RNA splicing [14,15]. hnRNPK drives the nuclear enrichment of some lncRNAs in human cells, and these lncRNAs were generally well spliced

[16]. However, the characteristics of the RNA secondary structures and lncRNA-mRNA interactions mediated by hnRNPK in vivo remain poorly understood. In this study, we discovered that hnRNPK selectively binds to small nuclear (sn)RNA U4, U11 and U12 and mediates most of the known secondary structures of them. However, we did not detect any binding with U1, U2, and U6, which are the primary components of spliceosomes. Additionally, we have identified hnRNPK-mediated interactions between lncRNAs and mRNA, which may play a role in regulating mRNA splicing.

## Results

### The general protocol of lhCLIP for detection of the in vivo RNA–RNA interactions mediated by hnRNPK

We performed lhCLIP experiments on HeLa cells, which involved the following steps: (1) HeLa cells are crosslinked with 0.3 J/cm$^2$ of UV-C (254 nm) irradiations and resuspended in cell lysis buffer; (2) HeLa cells lysate is immunoprecipitated with antibody-coupled beads; (3) protein-RNA complexes are cut with micrococcal nuclease randomly and dephosphorylated at their 3′ overhangs on beads; (4) the RNA 3′ ends are labelled with pCp–biotin and subsequently treated with FastAP alkaline phosphatase to remove the 3′ phosphate group from 'Cp–biotin', and the 5′ overhangs are phosphorylated with T4 polynucleotide kinase (PNK); (5) the resulting RNA fragments in close proximity are ligated; (6) Separation of samples into one-third and two-thirds parts, which are then resolved by SDS–PAGE using NuPAGE 4–12% Bis-Tris Gels respectively. After resolving RNP complexes, they were transferred onto nitrocellulose for further analysis (Fig 1A). One-third of the complexes were subjected to Chemiluminescent analysis, revealing that the pCp-biotin labeled hnRNPK-RNA complexes were distributed between 60 to 135 KDa (Fig 1B). The remaining two-thirds of the samples on nitrocellulose were used for total RNA extraction, which is then fragmented and converted into strand-specific libraries for sequencing (approximately 260 bp in length) (Figs 1A and S1C). The differences between lhCLIP, CRIC-seq, and other similar methods (CLASH, irCLASH, hiCLIP) are outlined in S1 and S2 Tables.

hnRNPK was detected as a 60 KDa band in the immunoprecipitation (IP) followed by Western blot assay using denatured SDS-PAGE gel electrophoresis (S1D Fig). During the lhCLIP experiment, conducted from step (1) to (5), we observed the presence of hnRNPK in both monomeric and dimeric forms using blue native gel electrophoresis, followed by blotting with an anti-hnRNPK antibody (Fig 1C). It's worth noting that both hnRNPK monomer and dimer require heating to eluate from the beads in the lhCLIP process. As a result, their electrophoretic behavior closely resembles that of denatured proteins/marker rather than the native protein marker. Moreover, it appears that the anti-hnRNPK antibody was co-eluted from the beads along with the hnRNPK protein and could not be separated during blue native gel electrophoresis. This led to the manifestation of a band at approximately 100 KDa for the hnRNPK monomer and its antibody complex. Similarly, the hnRNPK dimer, along with its antibody, was evident as a band around 160 KDa (Figs 1C and S1E). In contrast, the input sample, which was only UV cross-linked and then treated with loading buffer, showed a hnRNPK monomer band at around 60 KDa (Figs 1C and S1F). The hnRNPK dimer displayed a notably low abundance, posing challenges for its detection without IP enrichment in the input sample.

### lhCLIP captures known RNA-RNA interactions mediated by hnRNPK

The lhCLIP method demonstrated high reproducibility between biological replicates for both non-chimeric reads (R = 0.966, S1G Fig) and chimeric reads (R = 0.989) (Figs 1D, S1A, and

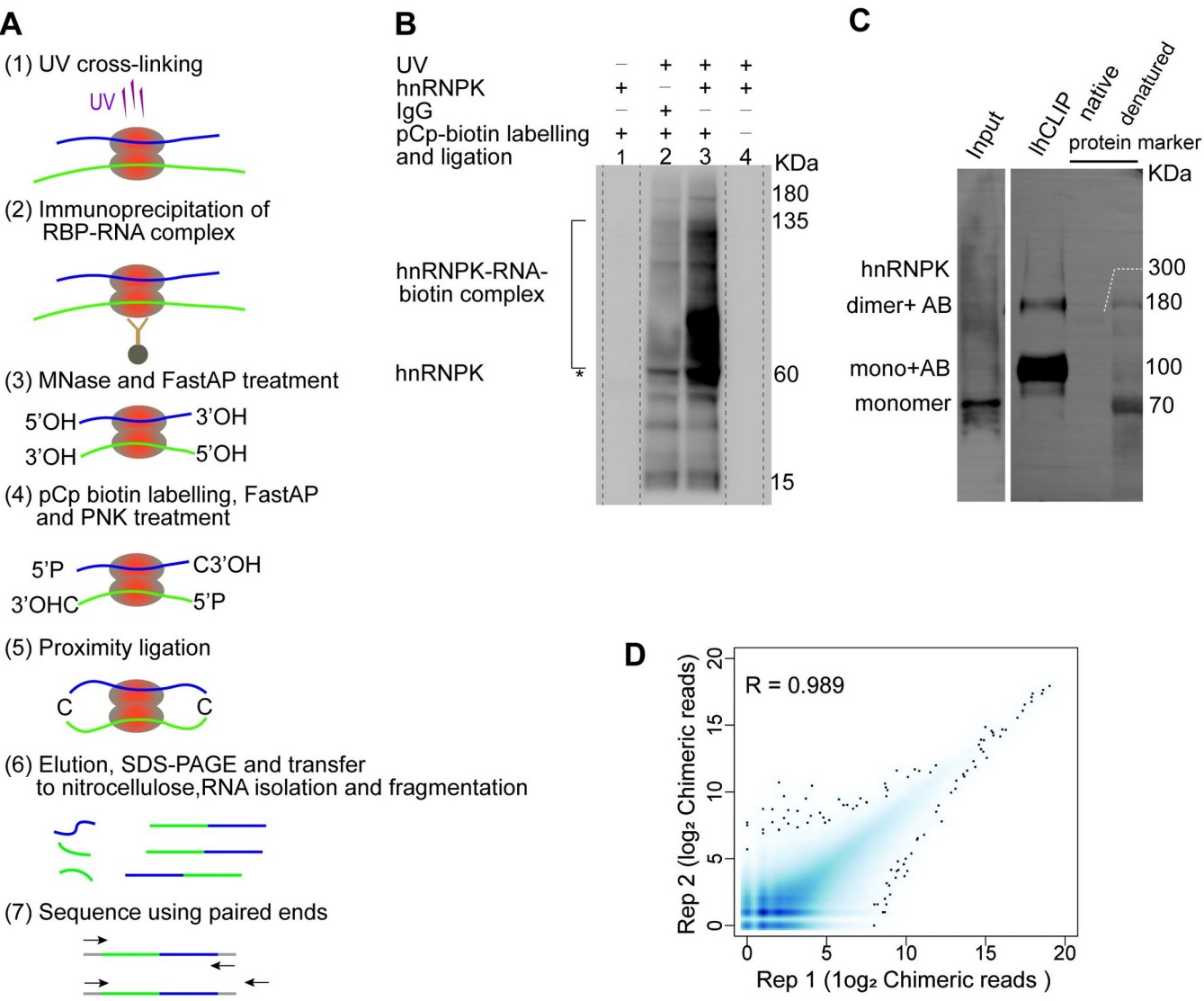

**Fig 1. Overview of lhCLIP.** (**A**) Schematic overview of the lhCLIP protocol. (**B**) Chemiluminescent analysis of the hnRNPK–RNA complexe that was isolated for the lhCLIP experiment. lhCLIP experiments were performed with anti-hnRNPK (lane 3), and the three controls in the absence of crosslinking (lane 1) or replaced anti-hnRNPK with IgG (lane 2) or omitted pCp biotin labelling (lane 4). (**C**) Analysis of the hnRNPK monomer and dimer captured in lhCLIP by blue native gel electrophoresis and blotting with anti-hnRNPK antibody. The pre-stained denatured protein marker was loaded in the right lane, with the native protein marker positioned adjacent to it. The white dotted line indicates the position of the 300 KDa native protein marker; however, it does not accurately correspond to the molecular weight of denatured hnRNPK. The hnRNPK monomer appears as a band around 60 KDa, while the hnRNPK monomer along with its antibody (AB) manifests as approximately 100 KDa. The hnRNPK dimer, together with its antibody (AB), is evident as a band around 160 KDa. (**D**) Correlation analysis of the chimeric reads between the replicates of lhCLIP.

S1B). We obtained 593850 chimeric reads (Fig 2A), which accounted for 4.87% of all uniquely mapped reads (S3 Table). We also observed a minor population of the chimeric reads in IgG samples, which represented the background of randomly ligated RNAs. However, less than 2% of the chimeric reads shared the same interactors between the hnRNPK and IgG samples (S2A Fig), demonstrating the specificity of RNA-RNA interactions detected by lhCLIP. The non-chimeric reads were comparable to traditional CLIP reads, with many of them mapped to the known hnRNPK binding RNAs, such as *MALAT1* and *CCAT1* (S2B Fig) [7]. Despite being present in relatively small amounts, hnRNPK binds to low expression RNAs, such as

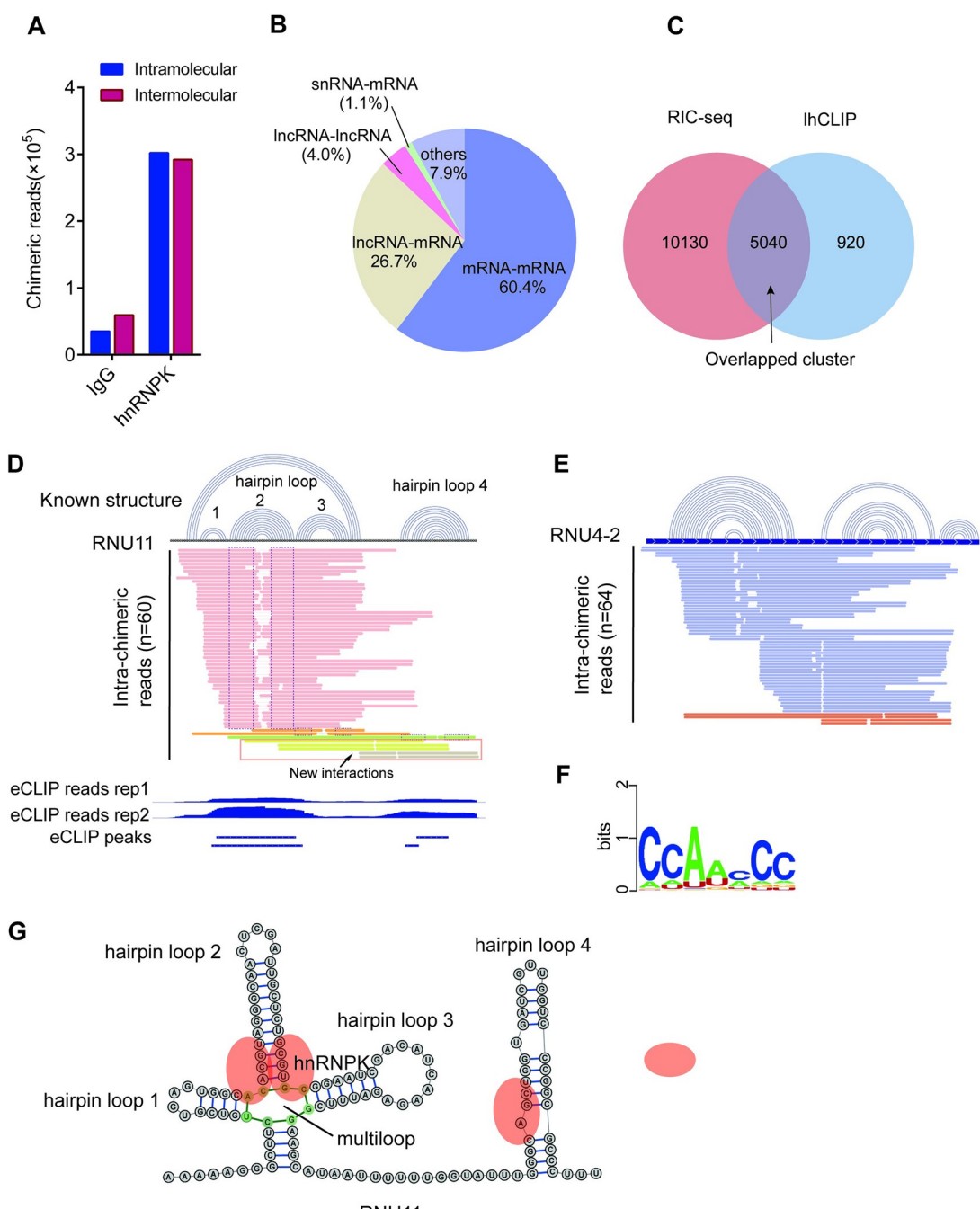

**Fig 2. lhCLIP identifies RNA interactions bound by hnRNPK.** (**A**) The numbers of intra and intermolecular chimeric reads.
(**B**) Distribution of intermolecular interactions among various classes of RNAs. (**C**) Venn diagram showing overlapping
duplexes detected by RIC-seq and lhCLIP methods. (**D**) Comparison of known structures of *RNU11* to that identified by lhCLIP.
Top, the know structure of *RNU11*. Middle, the intramolecular interactions of *RNU11* detected by lhCLIP. Intra-chimeric reads
in red box showing new interactions between the RNU11 hairpin loop 3 and 4 captured by lhCLIP. Bottom, eCLIP data of
hnRNPK in HepG2 cells showing hnRNPK binding sits on *RNU11*. (**E**) The known structure of RNU4-2 was compared to the
structure identified by lhCLIP. (**F**) The binding motif of hnRNPK was obtained from CISBP-RNA database. (**G**) Schematic
representations of the secondary structure of *RNU11* with the binding sits of hnRNPK.

*HSP90B2P*, which can be captured by lhCLIP. (S2C Fig). It has been previously reported that hnRNPK can form a dimer [7], which may contribute to the intermolecular RNA-RNA interactions. Of the 593850 chimeric reads, 291571 were found to be intermolecular chimeric reads (Fig 2A). The most enriched types of intermolecular interactions were mRNA-mRNA, lncRNA-mRNA, and lncRNA-lncRNA, while the proportion of snRNAs-mRNA interactions was only 1.1% (Fig 2B). For the high-confidence intramolecular interactions, multiple unique chimeric reads were assembled into a cluster for each transcript, with each cluster corresponding to one transcript. Significantly, 84.8% of the lhCLIP clusters were found to be part of the RNA-RNA interaction clusters that were identified by RIC-seq (Fig 2C), indicating that these RIC-seq clusters were mediated by hnRNPK.

The secondary structure of snRNA can affect RNA splicing by facilitating or inhibiting the binding of splicing factors to pre-mRNA [17,18]. We compared the known secondary structure of snRNA to that identified by lhCLIP and found that hnRNPK selectively binds to U11 (RNU11, Fig 2D), U4(RNU4-1and RNU4-2, Fig 2E) and U12 (S2D Fig), rather than snRNA U1, U2 and U6. This unexpected finding is supported by the hnRNPK eCLIP data [19]. For RNU11, lhCLIP captured most of the known intramolecular interactions (Fig 2D) [20]. However, the hairpin loop 1 structure of RNU11 was not captured by lhCLIP, as there was no hnRNPK binding in this region, which was confirmed by hnRNPK eCLIP binding peaks downloaded from ENCODE (Fig 2D). Besides the known intramolecular interactions, lhCLIP captured some new interactions between the RNU11 hairpin loop 3 and 4 (intra-chimeric reads in red box, Fig 2D). The previously reported binding motif of hnRNPK suggests a preference for binding to regions that are CA/U-rich [21] (Fig 2F). Furthermore, GraphProt analysis has indicated that hnRNPK tends to bind at multiloop regions [16,22], which is consistent with our data showing a high frequency of intramolecular reads captured at this region (Fig 2D). Accordingly, we illustrated the hnRNPK binding sites on RNU11 secondary structure (Fig 2G).

Small nucleolar (sno) RNAs are known to be involved in RNA splicing by modifying snRNAs [23]. Additionally, hnRNPK has been shown to bind to snoRNA [24]. We therefore compared the intramolecular interactions of *SNORD3B-1* and *SNORD3A* captured by lhCLIP with the published interaction structures [25,26]. Our analysis revealed similar intramolecular interactions between our data and the published data (S2E and S2F Fig) [25,26] indicating that lhCLIP captured the known intramolecular interactions of these snoRNA at hnRNPK binding regions.

## lhCLIP identified novel RNA-RNA interactome

To provide a comprehensive overview of RNA-RNA interactions mediated by hnRNPK and their characteristics, we constructed a transcriptome-wide RNA 3D interaction map using Juicebox [27]. Our analysis revealed extensive intra- and intermolecular RNA-RNA interactions on a transcriptomic scale (Figs 3A and S3A). The known hnRNPK binding lncRNAs, such as *NEAT1* [28,29] and *MALAT1* [30], showed considerable interactions (Fig 3A and 3B). In contrast, no *NEAT1-MALAT1* interaction was found from the IgG control (S3B Fig) or the control in which pCp-biotin labeling and ligation was omitted (S3C Fig). Additionally, a large number of RNAs, such as *PID1*, a growth-inhibitory gene in embryonal brain tumors and gliomas [31], were found to interact with mRNAs (Fig 3C and S4 Table). Notably, 16.4% of *MALAT1* target RNAs and 20.5% of *NEAT1* target RNAs in lhCLIP were among the RNA-RNA interactions revealed by RIC-seq (S3D Fig). *MALAT1* has previously been implicated in interacting with pre-mRNAs indirectly through protein intermediates [32]. In our study, we found that MALAT1 interacts with the intron of PID1 pre-mRNA in lhCLIP (S3E Fig), which was not

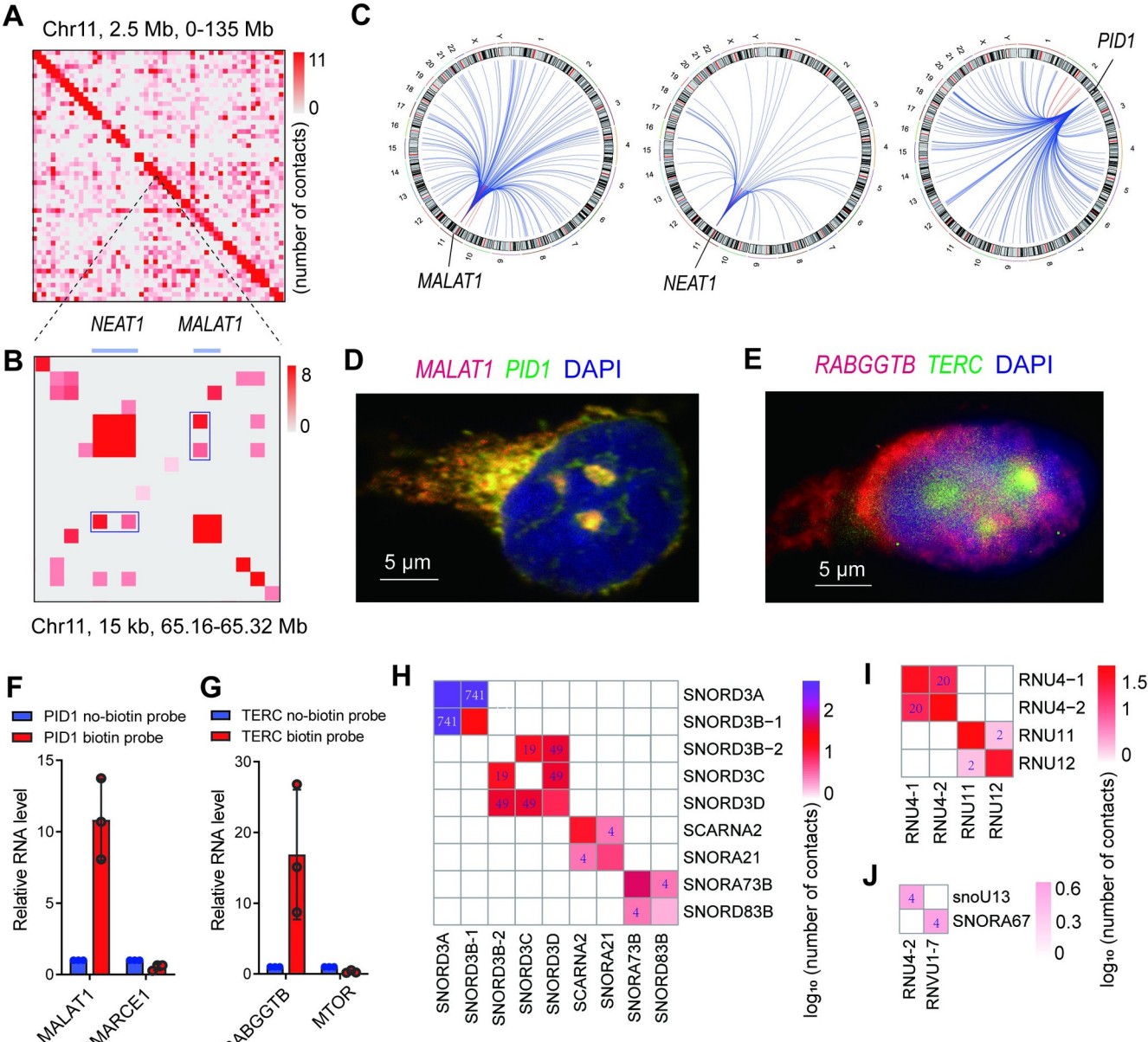

**Fig 3. lhCLIP identifies novel RNA-RNA interactome.** (**A**) RNA 3D map showing RNA-RNA interactions across chromosomes 11. (**B**) *NEAT1* and *MALAT1* interactions magnified from (**A**). Blue boxes denote the position of *NEAT1-MALAT1* interactions (**C**) Circos plot showing the *MALAT1-*, *NEAT1-* and *PID1*-interacting RNAs. The outer circles show different chromosomes, and the inner circles show unique contacts. (**D**) *MALAT1* and *PID1* pre-mRNA were colocalized by smFISH in HeLa cells. (**E**) RABGGTB and TERC RNA were colocalized by smFISH in HeLa cells. (**F**) RAP–qPCR validated that PID1 specificly binds with MALAT1 but not SMARCE1. Data are mean ± s.d.; n = 3 biological replicates. (**G**) RAP–qPCR validated that *TERC* specificly binds with *RABGGTB* but not *MTOR*. Data are mean ± s.d.; n = 3 biological replicates. (**H**) Heatmap of snoRNA-snoRNA interactions detected by lhCLIP, with the numbers of intermolecular contacts are shown in grids. The legend labels are $log_{10}$ (numbers of intermolecular contacts). (**I**) Heatmap of interactions among snRNA U4, U11 and U12 detected by lhCLIP. (**J**) Heatmap of snoRNA-snRNA interactions detected by lhCLIP.

detected by RIC-seq. To further investigate these newly discovered RNA-RNA interactions, we used single-molecule fluorescence in situ hybridization (smFISH). By using probes targeting distinct regions of MALAT1 and probes targeting the intron of PID1 pre-mRNA, we found that MALAT1 co-localizes with PID1 pre-mRNA in both the nucleus and cytoplasm (Figs 3D and S3F). It is worth to note that the localization of *MALAT1* in both nucleus and cytoplasm

was demonstrated in previous studies [33]. We further confirmed another interaction between lncRNA *TERC* and mRNA *RABGGTB* using smFISH (Fig 3E). RNA antisense purification (RAP) coupled with quantitative PCR (RAP–qPCR) was utilized to corroborate the interactions of *PID1* with *MALAT1* (Fig 3F) and *TERC* with *RABGGTB* (Fig 3G). In contrast, *SMARCE1* and *MTOR* were employed as negative controls, respectively, as there is no interaction between *SMARCE1* and *PID1*, or *MTOR* and *TERC*. In addition, lhCLIP reveals unexpected large numbers of snoRNA-snoRNA interactions that mediated by hnRNPK. Among the top six pairs of snoRNA-snoRNA interactions, the SNORD3A-SNORD3B-1 interaction had the highest frequency with 741 contacts (Fig 3H). In contrast, the interactions mediated by hnRNPK among snRNA U4, U11 and U12 were relatively low. lhCLIP did not detect interactions between U4-U11 and U4-U12, and only 2 contacts were identified between U11 and U12 (Fig 3I). Interactions between snoRNA-snRNA were very low, only two pairs of interactions were detected (Fig 3J). These results are consistent with the fact that interactions of snRNA-snRNA and snoRNA-snRNA are mainly RNA-duplexes but not proteins dependent [6]. We examined RNA- RNA interactions involving lowly expressed ncRNAs. Initially, we assessed the expression of RNAs in HeLa cell using RNA-seq data and selected three relatively lowly expressed ncRNAs: *SNHG12*, *HSP90B2P*, and *NKAPP1* (S2C Fig). Subsequently, we compiled a list detailing the inter-molecular RNA-RNA interactions involving the prominently expressed *NEAT1* transcript, alongside the lowly expressed transcripts *SNHG12*, *HSP90B2P*, and *NKAPP1*. Even though *NEAT1* interacts with a wide variety of diverse transcripts, it is surprising to discover that the minimally expressed *NKAPP1* demonstrates a contact count of 9 with *TMEM255A*, surpassing the majority of contact counts (ranging from 2 to 10) between *NEAT1* and its interacting transcripts (S6 Table). This observation challenges the notion that low-expressed RNAs consistently exhibit sparser interactions compared to their highly expressed counterparts. Taken together, lhCLIP can effectively capture both known and novel RNA-RNA interactions with high specificity.

## hnRNPK recruits a few ncRNA to alternative splicing events

To investigate whether the snRNA-mRNA and lncRNA-mRNA interactions mediated by hnRNPK participate in the regulation of RNA alternative splicing, we hypothesized that ncRNAs could regulate RNA alternative splicing only if they target at the RNA alternative splicing regions. To test this hypothesis, we analyzed the published RNA-seq data from hnRNPK knockdown and control HeLa cells using MISO to identify significant alternative splicing events [34]. We found that hnRNPK knockdown significantly altered 58 splicing events (S7 Table). After comparing the genomic positions of splicing events obtained from the MISO analysis with the RNA-RNA interaction sites detected by lhCLIP, we found that only one splicing event was bound with snRNA (*RNVU1-7*) (Fig 4A and S7 Table), four splicing events were bound with lncRNA (Figs 4B and S4A and S7 Table), which mediate by hnRNPK. Taken together, hnRNPK rarely recruited snRNAs or lncRNAs to the alternative splicing events. Since the alternative splicing event is regulated by multiple RBPs [35], hnRNPK is more likely to act cooperatively with other RBPs to control the assembly of the core splicing machinery and regulate alternative splicing indirectly. The precise mechanism that hnRNPK regulates alternative splicing need further research.

## Discussion

Our study has demonstrated that the lhCLIP method is capable of detecting both intra- and intermolecular RNA-RNA interactions associated with the specific protein of interest. This method not only recapitulates known RNA-RNA interactions, such as hnRNPK mediated

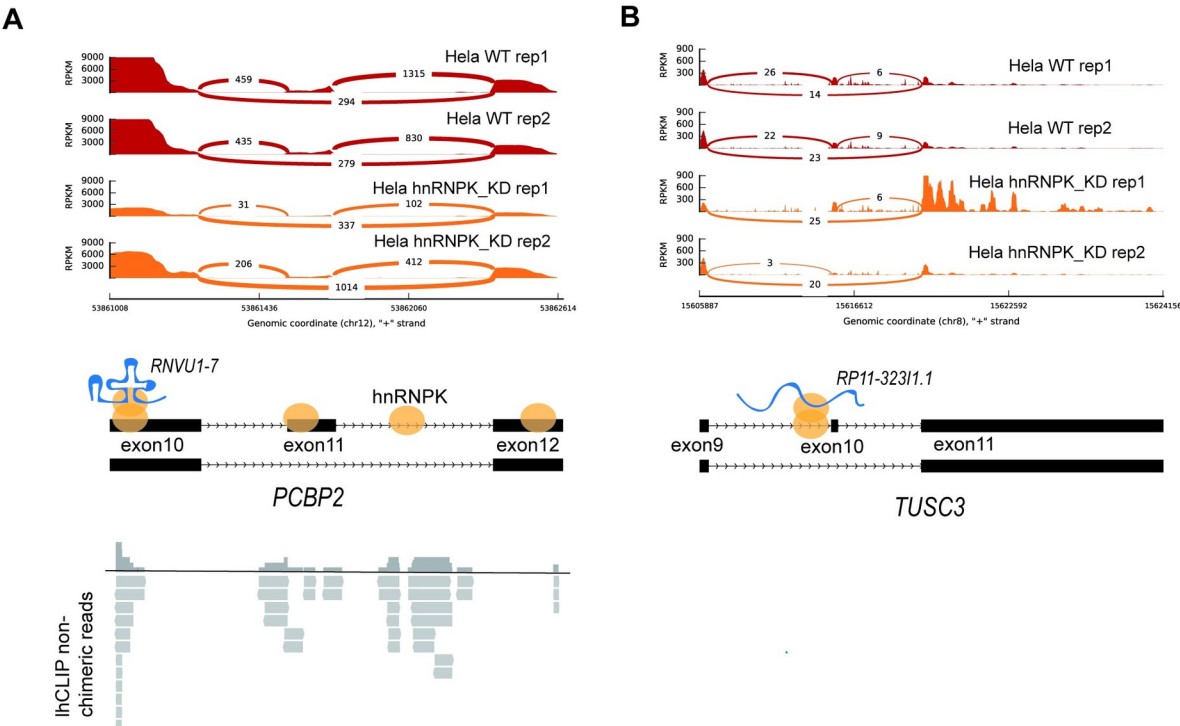

**Fig 4. lhCLIP identifies ncRNAs binding at RNA alternative splicing regions.** (**A**) snRNA *RNVU1-7* binds to RNA alternative splicing region of *PCBP2*, which is mediated by hnRNPK. Top, Sashimi plots illustrating alternative splicing event of *PCBP2* in WT HeLa cells and hnRNPK knock down cells. Middle, Schematic representations of *RNVU1-7—PCBP2* interaction in the RNA alternative splicing region mediated by hnRNPK. The orange oval represents the hnRNPK protein. Bottom, lhCLIP detected non-chimeric reads. (**B**) The lncRNA *RP11-323I1.1* binds to RNA alternative splicing region of *TUSC3*. Top, Sashimi plots illustrating alternative splicing event of *TUSC3* in WT HeLa cells and hnRNPK knock down ones. Bottom, Schematic representations of *RP11-323I1.1- TUSC3* interaction in the RNA alternative splicing region mediated by hnRNPK. The orange oval represents the hnRNPK protein.

snRNA secondary structure and *MALAT1-NEAT1* interaction, but also facilitates the generation of three-dimensional (3D) interaction maps of RNA and identification of novel RNA-RNA interactions mediated by specific protein in human cells.

Previous research has demonstrated that hnRNPK plays a role in regulating splicing [36]. In this study, we focused on the secondary structures of snRNA (RNU4, RNU11 and RNU12) and snoRNA (SNORD3A, SNORD3B-1) that are specifically bound by hnRNPK, and found that hnRNPK has a preference for binding at the stem loop regions of these RNAs. Furthermore, we observed that hnRNPK selectively binds with snRNA U4, which may be involved in regulating B complex aggregation and activation, ultimately affecting the process of RNA splicing [18]. Since U11 functions not only in recognizing the 5' splice site sequences, but also serves as an activator of U2-dependent alternative splicing and as a regulator of the U12-dependent spliceosome [37], hnRNPK may regulate RNA splicing of specific genes by binding with U11 and shaping its secondary structure.

LncRNA also participates in the regulation of RNA splicing. *MALAT1* depletion can affect alternative splicing [38] and reduce recruitment of Serine/arginine-rich (SR) proteins to active transcription sites [39]. This suggests that hnRNPK mediated *MALAT1*-preRNA interaction and influenced RNA processing, which may require further recruitment or modification of other proteins localized to these sites. Our research identified 58 RNA alternative splicing events regulated by hnRNPK, of which only 4 alternative splicing events were found to bind with lncRNA and one splicing event was found to bind with snRNA. This indicated that

hnRNPK may not directly recruit ncRNA to the specific alternative splicing events; Instead, hnRNPK may cooperate with multi-proteins and ncRNAs in the regulation of RNA alternative splicing. These RNA-RNA interactions that mediate by multi-proteins were not captured by lhCLIP. In other words, lhCLIP captures RNA-RNA interactions mediated directly by target protein.

Taken together, lhCLIP provides an accurate method to reveal RNA-RNA interactions mediated by a particular protein and the unique functions of the RNA binding protein. Our technique eliminates the use of radioisotopes while retaining the ability to visualize protein-RNA complexes, and does not require the introduction of exogenous expression of the target protein, thus broadening the application of lhCLIP-based assays in the scientific community.

## Limitations of the technology

The UV crosslinking used in lhCLIP selectively forms covalent bonds only at sites of direct protein-RNA interactions, leading to the exclusion of most indirect RNA-RNA interactions mediated by hnRNPK and other proteins from the analysis. As a result, the coverage of interacting RNAs remains insufficient, particularly for lowly expressed ncRNAs, which poses challenges in detecting their interactions. To address this limitation, researchers could explore options to enhance the experimental procedure or adopt alternative cross-linking methods, such as formaldehyde crosslinking or formaldehyde plus disuccinimidyl glutarate crosslinking.

## Methods

### lhCLIP libraries construction

**UV crosslinking.** HeLa (ATCC, CCL-2) cells were grown in MEM (Procell, PM150410) containing 10% fetal bovine serum and 0.5 mg/mL primocin (InvivoGen, ant-pm-1) on a 15-cm dish at 37 ˚C in 5% $CO_2$. For lhCLIP library construction, 3 × 15-cm dishes of HeLa cells were utilized. The cells were washed once by ice-cold PBS, and 4 ml ice-cold PBS was added. The cells were then subjected to 0.3 $J/cm^2$ of UV-C (254 nm) irradiations on ice, and collected by scraping. The cells from 3 dishes were transferred to a 15 mL tube and centrifugated at 500 g for 5 min at 4˚C, and the supernatant was removed.

**Preparation of antibody-coupled beads.** 100 μL Protein G Dynabeads (Thermo Fisher Scientific, 10004D) was blocked with 500 μL 5 mg/mL BSA/PBS at room temperature. Next, 10 μg anti-hnRNPK antibody (Santa Cruz, sc-28380) was conjugated to pre-blocked Protein G Dynabeads for 4 h at 4˚C in 200 μL cell lysis buffer (50 mM Tris-HCl pH 7.4, 100 mM NaCl, 1% NP-40 (Igepal CA630), 0.1% SDS, 0.5% sodium deoxycholate (protect from light)). Prior to immunoprecipitation, beads were washed twice with cell lysis buffer.

**Preparation of the cell lysate.** The cell pellet was resuspended in 1 mL cell lysis buffer Complete (cell lysis buffer supplemented with 1:100 Protease Inhibitor Cocktail (add freshly), and 10 μL SUPERaseIn (Life Technologies, AM2694)) and put on ice for 15 min. Next, 2 μL of Turbo DNase (LifeTech, AM2239) was added, and the sample was incubated in a Thermomixer at 1200 rpm and 37˚C for 5 minutes. After centrifuging at 15,000 g and 4˚C for 10 minutes, the supernatant was carefully transferred to a new tube.

**Immunoprecipitation.** The cell lysis buffer was removed from the beads, which were then added to the cell lysate. The bead/lysate mixture was incubated at 4˚C overnight with rotation. The supernatant was discarded using a magnetic stand and the beads were washed twice with high salt wash buffer (50 mM Tris-HCl pH 7.4, 1 M NaCl, 1 mM EDTA, 1% NP-40, 0.1% SDS, 0.5% sodium deoxycholate, the second wash was rotated for at least 5 min at 4˚C). The beads were then wash twice with 1 × PNK buffer (50 mM Tris-HCl pH 7.4, 10 mM $MgCl_2$, 0.2% NP-40).

**MNase digestion.**   The RNAs attached to the beads were digested with 6 U micrococcal nuclease (ThermoFisher, EN0181) in 200 μL of 1 × MN buffer (50 mM Tris-HCl pH 8.0, 5 mM CaCl$_2$) in an Eppendorf Thermomixer for 10 min at 37˚C, 15 s 1,400 r.p.m., 90 s rest. The reaction was stopped by 20 mM EGTA in 1 × PNK buffer.

**pCp–biotin labelling and proximity ligation.**   The RNA 5'- and 3'-terminal phosphate groups were removed with 10 U of FastAP alkaline phosphatase (ThermoFisher, EF0651) in an Eppendorf Thermomixer at 37 ˚C, 15 s 1,400 r.p.m., 90 s rest for 15 min. The RNA was then washed twice with 1 × PNK 20 mM EGTA buffer, twice with high salt wash buffer, and twice with 1 × PNK buffer. To label the RNA 3'-terminal, 2μL of 1 mM pCp–biotin (Thermo-Fisher, 20160) was added and the labelling was performed according to the manufacturer's instructions. After 3 times washing with 1 × PNK buffer, the phosphate groups of RNA·Cp–biotin were removed with 10 U of FastAP alkaline phosphatase. After thorough washing with 1 × PNK + 20 mM EGTA buffer, high salt wash buffer and 1 × PNK buffer successively, the RNA 5' ends were phosphorylated with T4 polynucleotide kinase (ThermoFisher, EK0032). After twice washing with 1 × PNK + EGTA buffer and twice 1 × PNK buffer (0.05% NP-40), the proximity ligation was accomplished with 50 U T4 RNA ligase (ThermoFisher, EL0021) in an Eppendorf Thermomixer overnight at 16˚C, 15 s 1,400 r.p.m., 90 s rest.

**SDS–PAGE and transfer of protein–RNA complexes to nitrocellulose.**   The next day, the RNAs bound to beads were washed 2 times with 1 × PNK buffer and resuspended in elution/loading master mix (20 μL wash buffer (20 mM Tris-HCl pH 7.4, 10 mM MgCl$_2$, 0.2% Tween-20), 7.5 μL of 4 × NuPAGE sample buffer, 3 μL 1 M DTT) and heated in a Thermo-mixer, 1200 rpm, 10 min at 70˚C. The samples were divided into one-third and two-thirds parts and respectively resolved using NuPAGE 4–12% Bis-Tris Gels (1.0 mm × 12 well) with SDS-PAGE at 160 V for 45 minutes. The resulting RNP complexes were then wet transferred onto nitrocellulose at 400 mA for 60 min at 4˚C.

**Chemiluminescent analysis of the pCp-biotin labeled hnRNPK–RNA complexes.**   The one-third of the samples that have been transferred onto a nitrocellulose membrane were analyzed using the LightShift Chemiluminescent RNA EMSA Kit (ThermoFisher, 20158) following the manufacturer's instructions. Briefly, Membrane was crosslink at 120mJ/cm$^2$ using a commercial UV-light crosslinking instrument equipped with 254nm bulbs. The membrane was carefully placed in a clean tray and blocked with 20mL of Blocking Buffer for 15 minutes with gentle shaking. After decanting the blocking buffer, the conjugate/blocking solution (66.7μL Stabilized Streptavidin-Horseradish Peroxidase Conjugate, 20 mL Blocking Buffer) was added to the membrane. The membrane was then incubated in the conjugate/blocking buffer solution for 15 minutes with gentle shaking. Next, the membrane was transferred to a new container and briefly rinsed with 20 mL of 1X wash solution. The membrane was washed four times for 5 minutes each in 20 mL of 1X wash solution with gentle shaking. After washing, the membrane was transferred to a new container and 30 mL of Substrate Equilibration Buffer was added. The membrane was incubated in the buffer for 5 minutes with gentle shaking. Carefully blotting an edge of the membrane on a paper towel to remove excess buffer, the membrane was then placed in a clean container and the Substrate Working Solution was poured onto the membrane so that it completely covered the surface. The membrane was incubated in the substrate solution for 5 minutes without shaking. Finally, the membrane was removed from the Working Solution and the hnRNPK-RNA complex was visualized using an ImageQuant Las4000mini.

**RNA isolation.**   The two-thirds of the samples that have been transferred onto a nitrocellulose membrane were placed on pre-wetted Whatman filter paper and the lane containing hnRNPK-RNA complex was cut using scalpels. Membranes were cut into 0.5–1 mm narrow strips that easily come to rest in the bottom of a siliconized 1.5-ml Eppendorf tube. 0.2 mL of

proteinase K reaction buffer (100 mM Tris, pH 7.5; 50 mM NaCl; 1 mM EDTA; 0.2% SDS) containing 10 µL of proteinase K (ThermoFisher Scientific, cat# AM2546) was added to each tube and incubated for 60 min at 50°C in an Eppendorf Thermomixer, 15 s 1,000 r.p.m., 30 s rest. After 60 min, tubes were removed from the thermomixer and briefly centrifuged. 200 µL of saturated-phenol-chloroform, pH, 6.7, was added to each tube and incubated for 10 min at 37°C in an Eppendorf Thermomixer, 1,400 r.p.m. Tubes were briefly centrifuged and the entire contents were transferred to a 2-mL Heavy Phase Lock Gel tube. After 2 min centrifugation at >13,000 r.p.m., the aqueous layer was re-extracted with 1 mL of chloroform (invert tube 10 times to mix; do not vortex, pipet, or shake) in the same 2-ml Heavy Phase Lock Gel tube and centrifuged for 2 min at >13,000 r.p.m., the aqueous layer was transferred to a siliconized 1.5-mL Eppendorf tube and precipitated overnight at −20°C by addition of 18 µl 5 M NaCl, 3 µL Linear Polyacrylamide (sigma, 56575), and 0.8 mL of 100% ethanol. The next day, RNA was pelleted at >13,000 r.p.m. for 45 min at 4°C, washed once with 1 mL of ice old 75% ethanol, and air dried. Pellets were resuspended in 8 µL RNase-free water.

**Strand-specific library preparation.** Strand-specific library was prepared with VAHTS Universal V8 RNA-seq Library Prep Kit for Illumina (Vazyme, NR605) according to manufacturer's instructions with several modifications. For fragmentation, 8 µL of total RNA was mixed with 8 µL of 2 × Frag/Prime Buffer and incubated at 94 °C for 5 min, then quickly put on ice. To synthesize the first-strand cDNA, 7 µL 1st Strand Buffer 3 and 2 µL1st Strand Enzyme Mix 3 was added to the RNA fragments and incubated at 25°C for 10 min, 42°C for 15 min, 70°C for 15 min, followed by 4°C Hold. We next created dUTP second-strand cDNA by adding the reaction mixture (2nd Strand Buffer 2 (5 µL with dNTP and 20 µL with dUTP), 15 µL 2nd Strand Enzyme Super Mix 2) to the first-strand cDNA and incubating at 16 °C for 30 min, 65°C for 15 min, 4°C Hold. Illumina Y-shaped adaptors were ligated to the dsDNA by adding the reaction mixture (25 µL Rapid Ligation Buffer 4, 5 µL Rapid DNA Ligase 4, 1 µL RNA Adapter, 4 µL $H_2O$) and incubated at 20°C for 15 min. The ligated DNA product were purified using 0.5 × VAHTS DNA Clean Beads (Vazyme, N411-01-AA) following the manufacturer's protocol and eluted in 20 µL of EB buffer. The adaptor-ligated cDNA was mixed with 1 µL of PCR Primer Mix 4, 1 µL of 50 mM MgSO4, 25 µL of 2 × HF Amplification Mix, and 3 µL of USER enzyme (NEB, M5505S). The mixture was incubated at 37 °C for 15 min to allow the USER enzyme to digest the dUTP strand, then at 94 °C for 2 min. PCR was performed with the following program: 98 °C for 45 s, 17 to 19 cycles of 98 °C for 15 s, 60 °C for 30 s and 72 °C for 30 s, final extension 72°C for 1 min. The 180–400-bp PCR products were selected using 1 × VAHTS DNA Clean Beads. The DNA concentration was measured by Qubit (ThermoFisher, Q32854) and sequenced using Illumina NovaSeq.

## Immunoprecipitation followed by western blot

For each IP, $5 \times 10^6$ HeLa were used. Cells was resuspended in 500 µL cell lysis buffer (50 mM Tris-HCl pH 7.4, 100 mM NaCl, 1% NP-40 (Igepal CA630), 0.1% SDS, 0.5% sodium deoxycholate (protect from light)) put on ice for 15 min. Extracts were cleared by centrifugation and incubated with anti-hnRNPK antibody (Santa Cruz, sc-28380) or mouse IgG Isotype control (R&D, catalog. no. MAB002), immobilized on protein G magnetic beads (Invitrogen) overnight at 4°C. Immunocomplexes were then washed with cell lysis buffer 3 times, resuspended in 30 µL 1 × NuPAGE sample buffer + reducing agent, incubated at 70°C for 10 minutes, and subjected to 4–12% SurePAGE precast gel (Genscript, M00656) and then transferred to nitrocellulose membranes (Millipore, HATF00010). After blocking with 5% BSA in Tris-buffered saline containing 0.1% Tween-20 (TBST) for 1 hour at room temperature, transferred membranes were incubated overnight at 4°C with primary anti-hnRNPK antibody (1:1000

dilution). After horseradish peroxidase-conjugated secondary antibodies incubation, bands were visualized using a ImageQuant Las4000mini.

## Blue native gel electrophoresis and immunoblotting

The lhCLIP sample processing involved UV crosslinking, immunoprecipitation, pCp-biotin labeling, ligation, and elution from beads using the elution/loading master mix (23 μL wash buffer (20 mM Tris-HCl pH 7.4, 10 mM MgCl2, 0.2% Tween-20) and 7.5 μL of 4 × NuPAGE sample buffer without reducing agents), followed by heating at 70°C for 10 minutes. The native protein marker (Beijing BioRab Technology, RFT264), the prestained protein marker (Thermo-Fisher, 26616), the lhCLIP sample, and the input sample (which underwent UV cross-linking and subsequent treatment with loading buffer) were then subjected to 4–13% Blue native PAGE gel (Beyotime Biotechnology, P0545S) and electrophoresed in cathode buffer B (50 mM Tricine, 7.5 mM Imidazole, 0.02% Coomassie blue G-250) at 100 V until blue running front has moved about one-third of the desired total running distance. The cathode buffer B was then replaced with cathode buffer B/10 (50 mM Tricine, 7.5 mM Imidazole, 0.002% Coomassie blue G-250) and electrophoresis was continued for another 1 h with the current limited to 15 mA. The gel was then soaked in Tris-glycine Western transfer buffer for 10 min and transferred to 0.45-μm PVDF. Filters were destained in methanol, washed in water, blocked, and probed with anti–hnRNPK antibody, followed by horseradish peroxidase-conjugated secondary antibodies.

## smFISH

Human *MALAT1* probes with Cy3 dye were purchased from RiboBio (lnc1100063), digoxigenin-labeled human *PID1* intron probes and *RABGGTB* probes, and biotin-labeled *TERC* probes were synthesized by Exonbio. RNA FISH was performed with D-T-G type mRNA in situ hybridization detection kit (Exonbio, D-00772) according to the manufacturer's instructions. The photographs were taken using a Nikon Eclipse Ni-E laser scanning confocal microscope.

## RAP-qPCR

RNA Antisense Purification (RAP) was performed with a RAP Kit (BersinBio, Bes5103-1) according to the supplier's protocol. Briefly, $2 \times 10^7$ cells were washed with PBS and cross-linked by 1% formaldehyde for 10 min with gentle agitation on a rotator at room temperature. Fixation was stopped by the addition of glycine (125 mM) and agitation for 5 min at room temperature. Fixed cells were washed twice in ice-cold PBS and lysed with 0.8mL lysis buffer. The cell lysate was mixed with an equal volume of 2 × Hybridization Buffer and denatured at 65°C for 10 minutes. Biotinylated antisense probes (40 pmol/probe) were denatured at 85°C for 3 minutes and added to the RAP system. Hybridization was carried out at 37°C for 30 minutes, followed by a 5-minute incubation at 50°C, and another round of hybridization at 37°C for 90–180 minutes. Next, 100 μL of streptavidin-coated magnetic beads were added and agitated for 30 min at room temperature. Non-specifically bound RNAs were removed by washing, and Trizol reagent was used to recover RNAs specifically pulled down by beads. RT-qPCR was used to analyze binding strength after reverse transcribing the RNAs. The probe and RT–qPCR primer sequences are listed in S5 Table.

## lhCLIP data processing and analysis

*Mapping and filtering of lhCLIP data* Because the structure of lhCLIP library is the same as that of RIC-seq, and both are used to obtain chimeric reads, lhCLIP data were processed according

to the protocol of RIC-seq data processing [40]. The analysis strategy is illustrated in S1A Fig. Adapters were trimmed off with the Trimmomatic program (v.0.36; trimmomatic PE -phred33 ILLUMINACLIP:adapter.fa:2:30:7:8:true LEADING:25 TRAILING:20 SLIDING-WINDOW:4:15 MINLEN:30) [41], and PCR duplicates were removed using homemade scripts developed by Zhaokui Cai et al [40]. The poly(N) tails at the 3′ ends were further clipped with the Cutadapt program (v.1.15; cutadapt -j 16 -b A{100} -b C{100} -b G{100} -b T{100} -n 3 -m 30 -e 0.1) [42]. After filtering, the paired reads were first aligned to 45S pre-rRNA, and the remaining reads were mapped to the human reference genome (assembly version: hg19) using the STAR software (v.020201) [43]. The parameters used were as follows: STAR—runMode alignReads—genomeDir index—readFilesIn read.fq—outFileNamePrefix outprefix—outFilterMultimapNmax 100—outSAMattributes All—alignIntronMin 1—score-GapNoncan -4—scoreGapATAC -4—chimSegmentMin 15 –chimJunctionOverhangMin 15—alignSJoverhangMin 15,—alignSJDBoverhangMin 10,—alignSJstitchMismatchNmax 5–1 5 5. STAR mapping produced normally mapped reads (Aligned_out.sam) and chimerically mapped reads (Chimeric_out.sam). The low-quality and secondary mapping results (mapping quality score < 30) were filtered using the SAMtools package (v.0.1.19) [44]. Gapped reads from normally mapped reads were extracted and reads resulting from RNA splicing were discarded. Chimerically mapped reads and remaining gapped reads constituted the set of chimeric reads for RNA structure and RNA-RNA interaction analysis. High-confidence intramolecular interactions were identified using the DG algorithm, requiring at least two unique chimeric reads with different termini and calculating a connection score (coverage A_B/$\sqrt{}$(coverage A × coverage B)) based on read coverage. A connection score cutoff of 0.01 was used to remove low-scoring RNA-RNA interactions. The intramolecular chimeric reads with different termini but overlapped at both arms can be clustered as high-confidence interactions if the connection score is >0.01. For intermolecular interactions, a Monte Carlo simulation was performed, comparing observed pairwise interactions to random ligations [45]. RNA–RNA interactions with local-background-corrected $P$ values lower than 0.05 were used for downstream analysis.

*Visualization of lhCLIP data in Integrative Genomics Viewer* Intramolecular chimeric reads were stored and sorted in a bam format file and displayed using the Integrative Genomics Viewer (IGV) visualization tool (v.2.14.0).

*Visualization of lhCLIP data in Juicebox* The Juicebox tool (v. 1.11.08) was applied to construct RNA–RNA interaction maps (http://aidenlab. org/juicebox). The chimeric read information was stored in a.short format file, in which each row recorded the genomic coordinates of two tags for a chimeric read. After sorting by chromosome names, this file was converted into.hic format using the pre command in Juicer tools [27] and further visualized as a heat map in Juicebox. In the two-dimensional heat maps, the colour intensities represented interaction counts.

## RNA-seq data analysis

RNA-seq data were extracted from previous report [46] (GSE217305) and analyzed with following steps. First, the sequencing quality of raw data were examined by package FastQC (v0.11.9). Then, the sequencing adapters were trimmed and read pairs with low quality were filtered from raw data through trimmomatic (v0.36) [41] with the following arguments: LEADING:30 TRAILING:30 SLIDINGWINDOW:4:15 MINLEN:50. The clean read pairs were mapping to reference genome (hg19) with STAR (v.020201) [43] with default arguments. Finally, the uniquely mapped reads were stored and sorted in a bam format file and displayed using the Integrative Genomics Viewer (IGV) visualization tool (v.2.14.0).

## eCLIP data analysis

In order to identify the binding sites of hnRNPK with *RNU11*, we downloaded the eCLIP binding reads and peaks identified by the ENCODE pipeline (accession: ENCFF147EIS ENCFF689WRD, ENCFF159LAW, ENCFF861DAK) from the ENCODE portal. This eCLIP data were displayed using IGV tool (v.2.14.0).

## Splicing analysis

The RNA-seq data of WT an hnRNPK knock down HeLa cell were downloaded from SRA database, accession SRP111756. HeLa cell RNA-seq data was analysed using MISO v.0.5.4 with default parameters comparing the hnRNPK knockdown samples to controls using differential splicing annotations obtained from the MISO website (miso_annotations_hg19_v2). After filtering for significant events (–num-inc 1 –num-exc 1–num-sum-inc-exc 10 –delta-psi 0.20 – bayes-factor 10), 58 splicing events were significant.

## Identifying alternative splicing events bound with ncRNA

The genomic positions of splicing events obtained from the MISO analysis were compared with RNA-RNA interaction sites detected by lhCLIP. A splicing event was considered as bound with an ncRNA if the exon or intron of the alternative splicing event bound with at least one ncRNA. ncRNAs which binding sites closest to alternative splicing events were also listed in S7 Table.

## Supporting information

**S1 Fig. Diagrams illustrating the mapping of chimeric reads and quality control of lhCLIP libraries.** (**A**) The mapping pipeline for lhCLIP data. (**B**) The relationship of intra- and inter-molecular chimeric reads to reference transcripts. (**C**) The lhCLIP libraries were quantified using a 2100 Bioanalyzer. (**D**) Western blot detected the immunoprecipitated hnRNPK. (**E**) Analysis of the hnRNPK monomer and dimer captured in IhCLIP by blue native gel electrophoresis. The gel image illustrates the Coomassie Blue G-250 stained complex of the hnRNPK monomer along with its antibody. It also displays a native protein mark at 300 KDa, alongside a visible denatured pre-stained protein marker ladder. (**F**) Analysis of IhCLIP input sample by blue native gel electrophoresis and blotting with anti-hnRNPK antibody. In the left panel, the bands developed through ECL are displayed, while the right panel exhibits a denatured protein marker ladder captured under white light. Notably, the hnRNPK monomer band is evident around 60 KDa. (**G**) Correlation analysis of the non-chimeric reads between the replicates of lhCLIP.
(TIF)

**S2 Fig. lhCLIP identifies RNAs bound by hnRNPK.** (**A**) The percentage of chimeric reads that are shared between the IgG and hnRNPK samples. (**B**) hnRNPK binding sites at *MALAT1* and *CCAT1* RNA revealed by lhCLIP. (**C**) Comparing the binding signals of hnRNPK with ncRNAs expressed at different levels. Up, The IGV shows the *NEAT1*, *SNHG12*, *HSP90B2P*, and *NKAPP1* RNA-seq reads. Bottom, The IGV shows the *NEAT1*, *SNHG12*, *HSP90B2P*, and *NKAPP1* lhCLIP non-chimeric reads. (**D**), (**E**) and (**F**) Comparison of intramolecular interactions of RNU12 (**D**), *SNORD3B-1* (**E**) and *SNORD3A* (**F**) identified by lhCLIP to their know structure.
(TIF)

**S3 Fig. lhCLIP identified novel RNA-RNA interactome.** (**A**) RNA 3D map showing RNA-RNA interactions across all chromosomes. (**B**) *No NEAT1-MALAT1 interaction was* detected from IgG sample. hnRNPK sample is shown at the bottom left of the Fig. IgG sample is shown at the top right of the Fig. The blue boxes denote the position of *NEAT1-MALAT1* interactions (**C**) No *NEAT1-MALAT1* interaction was detected from control in which pCp-biotin labeling and ligation was omitted. hnRNPK sample is shown at the bottom left of the Fig. Control sample is shown at the top right of the Fig. The blue boxes denote the position of *NEAT1-MALAT1* interactions (**D**) Overlaps of *MALAT1* and *NEAT1* interacting sites identified by lhCLIP and RIC-seq. Hypergeometric test was used to calculate the *P* value. (**E**) Positions of hybridization *PID1* probes labelled with digoxigenin. (**F**) The colocalization of *MALAT1* and *PID1* pre-mRNA in HeLa cells was detected by smFISH.
(TIF)

**S4 Fig.** (**A**) lncRNA *RP11-118E18.4* binds to the RNA alternative splicing region of *CCT4*. Top, Sashimi plots illustrating alternative splicing event of *CCT4* in WT HeLa cells and hnRNPK knock down cells. Bottom, Schematic representations of *RP11-118E18.4—CCT4* interaction in the RNA alternative splicing region mediated by hnRNPK.
(TIF)

**S1 Table. The difference between lhCLIP and CRIC-seq.**
(DOCX)

**S2 Table. Comparative analysis of various techniques for mapping RNA interactions mediated by a specific RBP.**
(DOCX)

**S3 Table. The mapping results of lhCLIP libraries.**
(XLSX)

**S4 Table. List of *MALAT1*, *PID1*and *NEAT1* interacting RNAs in vivo.**
(XLSX)

**S5 Table. The sequence of qPCR primers and RAP-qPCR probes.**
(XLSX)

**S6 Table. List of *NEAT1*, *SNHG12*, *HSP90B2P* and *NKAPP1* interacting RNAs and number of contacts in vivo.**
(XLSX)

**S7 Table. Significant RNA alternative splicing events affected by hnRNPK knockdown, and RNA-RNA interaction of these RNA.**
(XLSX)

## Author Contributions

**Conceptualization:** Yuanlang Hu, Yujuan Fan, Shiqi Jia.

**Data curation:** Yuanlang Hu, Tao Hao, Hanwen Yu.

**Formal analysis:** Yuanlang Hu, Tao Hao, Hanwen Yu.

**Funding acquisition:** Yuanlang Hu, Weihua Tao, Jichang Wang, Shiqi Jia.

**Investigation:** Yuanlang Hu, Tao Hao, Hanwen Yu, Wenbin Miao, Yi Zheng, Weihua Tao, Jingshen Zhuang.

**Methodology:** Yuanlang Hu, Shiqi Jia.

**Resources:** Yuanlang Hu, Jichang Wang, Yujuan Fan, Shiqi Jia.

**Supervision:** Yujuan Fan, Shiqi Jia.

**Validation:** Yuanlang Hu, Tao Hao.

**Visualization:** Yuanlang Hu, Tao Hao, Hanwen Yu.

**Writing – original draft:** Yuanlang Hu.

**Writing – review & editing:** Yuanlang Hu, Jichang Wang, Yujuan Fan, Shiqi Jia.

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
