## [Decision Letter · Decision Letter 0]

11 Jun 2023

Dear Dr Jia,

Thank you very much for submitting your Methods entitled 'lhCLIP reveals the in vivo RNA–RNA interactions recognized by hnRNPK' to PLOS Genetics.

The manuscript was fully evaluated at the editorial level and by independent peer reviewers. The reviewers appreciated the attention to an important problem, but raised some substantial concerns about the current manuscript. Based on the reviews, we will not be able to accept this version of the manuscript, but we would be willing to review a much-revised version. 

If you decide to revise the manuscript for further consideration at PLOS Genetics, please aim to resubmit within the next 60 days, unless it will take extra time to address the concerns of the reviewers, in which case we would appreciate an expected resubmission date by email to plosgenetics@plos.org.

We are sorry that we cannot be more positive about your manuscript at this stage. Please do not hesitate to contact us if you have any concerns or questions.

Yours sincerely,

Guanzheng Luo

Guest Editor

PLOS Genetics

Quanjiang Ji

Section Editor

PLOS Genetics

Reviewer's Responses to Questions

**Comments to the Authors:**

Reviewer #1: Hu et al. have established lhCLIP technology to analyze the RNA-RNA interactions mediated by a specific RNA binding protein (RBP), which provides a useful approach for investigating the roles of specific RBP in RNA metabolism. This approach shows its advances in omitting the application of radioisotopes and detecting only the direct interactions of RNA to specific RBP (crosslinked by UV light). However, two small points need to be clarified to improve the manuscript for publication in PLoS Genetics.

1. The authors claimed that dimerization of HNRNPK could explain the intermolecular RNA-RNA interactions mediated by hnRNPK. However, the co-IP result (Fig S1C) showed that dimerized HNRNPK are barely co-precipitated with the procedure. If the WB was run under denatured condition, the authors at least ran a gel with native condition (similar in Fig 1B) to show that the dimer of hnRNPK was indeed captured in IhCLIP.

2. Why did the authors not detect the interaction of TERC with RABGGTB using smFISH? Or using RAP-qPCR for PID1 with MALAT1 to confirm the interaction?

Reviewer #2: In this manuscript, the authors developed a novel method for capturing RNA-RNA interactions mediated by a specific RBP, named as lhCLIP. It’s useful and helpful for other researchers, but there are some questions should be addressed before the manuscript can be published:

1. The authors said that the structure of ilCLIP library is the same as that of RIC-seq, while the authors of RIC-seq developed a CRIC-seq (https://doi.org/10.1016/j.molcel.2023.03.001, published in 2023) depending on RIC-seq, and CRIC-seq can capture RNA-RNA interactions mediated by a specific RBP, so the authors should present the difference between ilCLIP and CRIC-seq.

2. While ilCLIP is a new method, the authors should compare ilCLIP with the other similar methods (CLASH, hiCLIP, irCLASH, CRIC-seq, and so on) in detail, for example, the library time, the chimeric rate, and so on.

3. In the manuscript, the authors said that both CLASH and hiCLIP require overexpression of RBP, how can the authors get this conclusion? Sometimes, while there is no effective antibody of target RBP, or the expression of target RBP is low in target cell line, maybe some other causes, the author may overexpress the target RBP with FLAG or GFP label and use the label’s antibody to capture the target RBP’s RNA substrates, but that does not mean that both these methods require overexpression. At the same time, the authors said lhCLIP does not require the introduction of any exogenous protein-coding genes, so while there is not the target RBP’s antibody, how can you using lhCLIP to capture the target RBP’s RNA substrates?

4. The authors should present the detailed messages for their analysis, for example, how to annotate the chimeric or non-chimeric reads, how to define the cluster in Figure 2C, and so on; the detailed parameters for each software; for some figures, the authors should write the figure legend in detail, for example, the Figure1B.

5. There are some wrong spelling, for example, in Figure 3A, should be “contacts”, in line 193, should be “number”, and so on.

Reviewer #3: The manuscript “IhCLIP reveals the in vivo RNA-RNA interactions recognized by hnRNPK”, Hu et al., developed IhCLIP that can investigate RNA-RNA interactions and the mediated specific proteins with UV crosslinking and proximity ligation and immunoprecipitation techniques. Big advantage of the method is that it provides both information of RNA-RNA and mediated specific protein complex at same time. I believe that the IhCLIP is important for further interactome analysis study, however I have several comments that authors should address.

In this study, authors selected hnRNPK to investigate their IhCLIP technology. hnRNPK is one of most famous RNA binding proteins that is known to bind many types of RNAs for splicing function, nucleocytoplasmic function and mRNA translation. I wonder whether the technology can be broadly used for other RBPs.

In addition, authors focus on very highly expressed nuclear transcripts such as NEAT1, MALAT1, snRNA, snoRNA etc. in this study. I would imagine that capturing and sequencing of hnRNPK binding on such RNAs are much easier than lowly expressed ncRNAs and lncRNAs. Authors must investigate lowly expressed RNA-RNA interactions and mention the limitation of the technology.

The advantage of this method is that the RNA-RNA interactions are captured after crosslinking and proximity ligation. So that we can capture many RNAs that are not only directly bind hnRNPK but also bind hnRNPK binding proteins. However, I am afraid if we can understand whole interacted molecules of such splicing complexes with this technology because other proteins than hnRNPK are missing from the analysis. Authors should discuss and suggest further possible analysis or validation of interactome study with the technology.

CLIP technologies are depending on antibody affinity to target protein to capture protein-RNA complexes. Even though we select two different types of antibodies to same target protein, results can be different with different antibodies. Can authors suggest how we can select good and proper antibodies for the specific results of IhCLIP and how we can understand that the results are not false positive?

**Have all data underlying the figures and results presented in the manuscript been provided?**

Reviewer #1: None

Reviewer #2: None

Reviewer #3: Yes

PLOS authors have the option to publish the peer review history of their article (what does this mean?). If published, this will include your full peer review and any attached files.

Reviewer #1: No

Reviewer #2: **Yes: **Yulong Song

Reviewer #3: No

---

## [Decision Letter · Decision Letter 1]

5 Oct 2023

Dear Dr Jia,

We are pleased to inform you that your manuscript entitled "lhCLIP reveals the in vivo RNA–RNA interactions recognized by hnRNPK" has been editorially accepted for publication in PLOS Genetics. Congratulations!

Yours sincerely,

Guanzheng Luo

Guest Editor

PLOS Genetics

Quanjiang Ji

Section Editor

PLOS Genetics

Comments from the reviewers (if applicable):

Reviewer's Responses to Questions

**Comments to the Authors:**

Reviewer #1: The authors have clarified all my concerns and I support its publication in PLOS Genetics.

Reviewer #2: The authors have addressed my questions.

Reviewer #3: The authors addressed all my concerns in the manuscript. I believe that the work should be published.

**Have all data underlying the figures and results presented in the manuscript been provided?**

Reviewer #1: None

Reviewer #2: None

Reviewer #3: Yes

PLOS authors have the option to publish the peer review history of their article (what does this mean?). If published, this will include your full peer review and any attached files.

Reviewer #1: No

Reviewer #2: No

Reviewer #3: No

**Data Deposition**

http://datadryad.org/submit?journalID=pgenetics&manu=PGENETICS-D-23-00429R1

**Press Queries**

---

## [Editor Report · Acceptance letter]

13 Oct 2023

PGENETICS-D-23-00429R1 

 lhCLIP reveals the in vivo RNA–RNA interactions recognized by hnRNPK 

Dear Dr Jia, 

We are pleased to inform you that your manuscript entitled " lhCLIP reveals the in vivo RNA–RNA interactions recognized by hnRNPK " has been formally accepted for publication in PLOS Genetics! Your manuscript is now with our production department and you will be notified of the publication date in due course.

With kind regards,

Anita Estes

PLOS Genetics

On behalf of:
